# Potential Effects of Indole-3-Lactic Acid, a Metabolite of Human Bifidobacteria, on NGF-Induced Neurite Outgrowth in PC12 Cells

**DOI:** 10.3390/microorganisms8030398

**Published:** 2020-03-12

**Authors:** Chyn Boon Wong, Azusa Tanaka, Tetsuya Kuhara, Jin-zhong Xiao

**Affiliations:** Next Generation Science Institute, Morinaga Milk Industry Co., Ltd., Zama 252-8583, Japan; chynboon020@morinagamilk.co.jp (C.B.W.); ad-tanaka@morinagamilk.co.jp (A.T.); t_kuhara@morinagamilk.co.jp (T.K.)

**Keywords:** tryptophan metabolites, indole-3-lactic acid, *Bifidobacterium*, neurite outgrowth

## Abstract

Gut microbiota-derived tryptophan metabolites such as indole derivatives are an integral part of host metabolome that could mediate gut–brain communication and contribute to host homeostasis. We previously reported that infant-type Human-Residential Bifidobacteria (HRB) produced higher levels of indole-3-lactic acid (ILA), suggesting the former might play a specific role in microbiota–host crosstalk by producing ILA in human infants. Nonetheless, the biological meaning of bifidobacteria-derived ILA in infant health development remains obscure. Here, we sought to explore the potential role of ILA in neuronal differentiation. We examined the neurite outgrowth and acetylcholinesterase (AchE) activity of PC12 cells following exposure to ILA and NGF induction. We found that ILA substantially enhanced NGF-induced neurite outgrowth of PC12 cells in a dose-dependent manner, and had the most prominent effect at 100 nM. Significant increases in the expression of TrkA receptor, ERK1/2 and CREB were observed in ILA-treated PC12 cells, suggesting ILA potentiated NGF-induced neurite outgrowth through the Ras/ERK pathway. Additionally, ILA was found to act as the aryl hydrocarbon receptor (AhR) agonist and evoked NGF-induced neurite outgrowth in an AhR-mediated manner. These new findings provide clues into the potential involvement of ILA as the mediator in bifidobacterial host–microbiota crosstalk and neuronal developmental processes.

## 1. Introduction

The human gastrointestinal tract harbors a diverse and dynamic microbial community called the gut microbiota, that plays a vital role in human biology and health, including metabolic and nutritional homeostasis, immune system maturation and stimulation and even brain activity [1,2]. The symbiotic relationship between the host and a myriad of microorganisms relies on complex molecular crosstalk, with microbial metabolites acting as major mediators [3]. Recent advances in microbial metabolomics studies have identified a few gut bacterial metabolites derived from specific dietary components at the interface between the host and its microbiome, one of which is the tryptophan metabolites [4].

Tryptophan is an essential amino acid bearing an indole ring, derived from dietary proteins [5]. It is mainly digested and absorbed in the small intestine, but significant amounts of tryptophan may persist to the colon [6], where they are metabolized by the gut bacteria resulting in a variety of indole derivatives [4]. Apart from its role in protein synthesis, tryptophan and its metabolites such as indole derivatives have increasingly been recognized as essential orchestrators of host physiology, and may contribute to intestinal and systemic homeostasis [2,3,4,7]. New research suggests that microbial tryptophan metabolites may also have a positive impact on inflammatory responses [8] and neurological functions [9].

Evidence is accumulating that microbial tryptophan-derived indole compounds, including indole-3-acetic acid (IAA), indole-3-aldehyde (IAld), indole-3-lactic acid (ILA) and indole-3-propionic acid (IPA) act as ligands of the aryl hydrocarbon receptor (AhR) [10,11]. These indole derivatives may potentially improve intestinal barrier function [12], regulate the gut mucosal immune system [13] and modulate inflammatory responses [14] in an AhR-dependent way. Studies have also been reported that these tryptophan metabolites can inhibit amyloid fibrillation of lysozymes, and that they possess neuroprotective properties [15]. For instance, ILA and IPA have been identified as antioxidants and free-radical scavengers [16,17,18]. IPA was found to exhibit potent neuroprotective effects against the Alzheimer’s β-amyloid [17], as well as neuronal damage and oxidative stress in the ischemic hippocampus [9].

In a recent report, we found that ILA was the only tryptophan metabolite produced by bifidobacterial species wherein infant-type Human-Residential Bifidobacteria (infant-type HRB) produced higher levels of ILA than those species inhabiting the adult and animal intestine [19]. Numerous studies have also demonstrated that the predominant bifidobacterial species in the infant gut microbiome produce ILA [20,21,22], suggesting that bifidobacteria-derived ILA may have conserved roles in infant health development. New research has reported that ILA produced by an infant-type HRB strain *B. longum* subsp. *infantis* ATCC 15697 interacts with AhR and reduces the inflammatory interleukin (IL)-8 response in immature but not mature intestinal enterocytes [23]. Very recently, in a preprint, the concentration of ILA in the fecal samples of infants obtained from birth until 6 months of age was found to be positively correlated with the abundance of breast milk-promoted *Bifidobacterium* species harboring the aromatic lactate dehydrogenase enzyme, suggesting that ILA could be a relevant early life AhR agonist and could be associated with intestinal and systemic homeostasis [24]. Combined with the fact that indole derivatives could modulate intestinal and systemic homeostasis and exert a neuroprotective effect, this makes bifidobacteria-derived ILA in early life neuronal development a very relevant area of research, which has not received much attention.

Here, we sought to explore the potential role of ILA, which is the key metabolite of infant-type HRB, in neuronal differentiation. The rat adrenal pheochromocytoma (PC12) cell line, which represents a well-established model system for the investigation of neuronal differentiation and function [25,26], was used to elucidate the effects of ILA on neuritogenesis, following nerve growth factor (NGF) induction. We examined neurite outgrowth and acetylcholinesterase (AchE) activity, which are known differentiation phenotypes of PC12 cells [25,27], following exposure to ILA and NGF induction. Interestingly, we found that ILA significantly enhanced NGF-induced neurite outgrowth of PC12 cells in a dose-dependent manner. We report here for the first time that the indole derivative, ILA, could potentially promote NGF-induced neuronal differentiation via the Ras/ERK pathway, as well as in an AhR-dependent way.

## 2. Materials and Methods

### 2.1. Test Compounds and Reagents

ILA was purchased from Tokyo Chemical Industry Co., Ltd. (Tokyo, Japan). Tryptophan, papaverine hydrochloride, α-naphthoflavone (AFN) and CH223191, as well as other chemicals, were purchased from Sigma-Aldrich (St. Louis, MO, USA) unless otherwise stated. Dimethyl sulfoxide (DMSO) and methanol were obtained from Wako (Osaka, Japan). ANF and CH223191 were dissolved at the concentration of 10 mM in DMSO. The stock solutions were serially diluted in sterile MiliQ water to prepare the analytical samples. Nerve growth factor (NGF; 2.5S) was purchased from Alomone Labs (Jerusalem, Israel).

### 2.2. Cell Culture

PC12 cell, a rat adrenal pheochromocytoma cell line, was purchased from the European Collection of Authenticated Cell Cultures (ECACC 88022401; Salisbury, UK). The floating cells were maintained in Roswell Park Memorial Institute 1640 medium (RPMI; Gibco Life Technologies, Grand Island, NY, USA) supplemented with 10% (*v/v*) heat-inactivated horse serum (HS; Gibco Life Technologies, NY, USA), 5% (*v/v*) fetal bovine serum (FBS; Gibco Life Technologies, NY, USA), and 0.1% (*v/v*) penicillin/streptomycin (Gibco Life Technologies, NY, USA) in an atmosphere of 5% CO_2_ at 37 °C. The medium was replaced every three days.

### 2.3. Dose-Response of ILA

PC12 cells (passage number < 13) were seeded in collagen type IV-coated 24-well culture plates (Iwaki, Shizuoka, Japan) at a density of 10,000 cells/mL per well with the complete growth medium (RPMI supplemented with 10% (*v/v*) HS, 5% (*v/v*) FBS, and 1% (*v/v*) penicillin/streptomycin) for 24 h. The cells were changed to low serum (1% *v/v* HS) medium for 24 h. After that, the cells were treated with NGF (25 ng/mL) and ILA at a wide range of final concentrations (1 µM, 100 nM, 10 nM, and 1 nM) for five consecutive days. The low serum medium and test compounds were replaced on day 3. Papaverine hydrochloride and tryptophan added under the same conditions were used as a reference control [28] and a baseline compound, respectively. Nontreated control (cells without NGF or any test compound) and NGF control (cells treated with NGF only) were also grown under the same conditions. The cells were then subjected to immunofluorescence staining for quantification of neurite outgrowth and acetylcholinesterase assay. After that, the maximally effective concentration of ILA was selected for further analyses.

#### 2.3.1. Immunofluorescence Staining

Following treatment for five days, cells in transparent collagen IV-coated plates were washed twice with phosphate-buffered saline (PBS) at room temperature and fixed for 5 min using absolute methanol (precooled to -20 °C). Cells were washed thrice with ice-cold PBS and incubated with blocking buffer PBST (1% (*w*/*v*) bovine serum albumin/10% (*v/v*) normal goat serum/0.3 M glycine in PBS containing 0.1% (*v/v*) Tween-20) for 1 h. Cell body and processes were then labeled with an anti-βIII-tubulin mouse primary antibody (1 µg/mL; Abcam, Cambridge, UK) diluted in PBST containing 1% (w/v) bovine serum albumin at 4 °C overnight, followed by an Alexa Fluor 488-conjugated goat anti-mouse secondary antibody (2 µg/mL; Abcam, Cambridge, UK) for 1 h at room temperature, protected from light. For nuclear staining, cells were counterstained with Cellstain^®^ 4′,6′-Diamidino-2-phenylindole (DAPI; Dojindo Molecular Technologies, Kumamoto, Japan) for 10 min at 37 °C. Plates were then subjected to image acquisition and analysis of neurite outgrowth.

#### 2.3.2. Quantification of Neurite Outgrowth

Plates containing fluorescent-labeled cells were examined under an inverted fluorescence microscope (DP73; Olympus, Tokyo, Japan) using a multiple bandpass emission filter and matched excitation filters for the blue channel (nuclei) and green channel (cell body and processes) at a magnification of 100x. Four images per well were captured. Cells displaying projections at least 1.5 times longer than the length of the cell body were considered positive and counted as neurite-bearing cells. Counting was performed in a blinded manner. The percentage of neurite-bearing cells was calculated as the percentage of the number of neurites divided by the total number of cells. Each data point corresponds to the counts obtained from three independent wells.

#### 2.3.3. Analysis of Acetylcholinesterase (AchE) Activity

The thiocholine produced from the hydrolysis of acetylthiocholine by the endogenous AchE enzyme in each sample was quantified by a fluorescence colorimetric assay to examine the effect of ILA on AchE activity, which is a biochemical marker for neuronal differentiation in PC12 cells [28]. In brief, PC12 cells were grown and treated for five consecutive days, as mentioned. The cells were then lysed with ice-cold NP-40 cell lysis buffer containing 150 mM NaCl, 50 mM Tris (pH 8.0), 2 mM EDTA (pH 8.0) and 1% (*v/v*) NP-40. AchE activity of the cell lysates was determined by the Amplite Fluorimetric Acetylcholinesterase Assay Kit (AAT Bioquest, Sunnyvale, CA, USA) according to the manufacturer’s instructions. Assay signals were read with a fluorescence absorbance microplate reader (SH-9000, Corona Electric, Ibaraki, Japan). at Ex/Em = 490/520 nm. The AchE activity was determined from the standard curve and normalized with the protein concentration in each sample. The protein concentration was determined by Pierce bicinchoninic acid (BCA) protein assay kit (Invitrogen, Paisley, UK) with bovine serum albumin as a standard.

### 2.4. Western Blot Analysis of Phospho-TrkA, ERK, Phospho-ERK, CREB and Phospho-CREB Proteins

PC12 cells (passage number < 13) were seeded in collagen type IV-coated 24-well culture plates at a density of 50,000 cells/mL per well with the complete growth medium for 24 h, before being shifted to a low serum (1% *v/v* HS) medium for 24 h before exposure to NGF (25 ng/mL) and the test compounds (reference control, ILA or tryptophan at a final concentration of 100 nM) for 24 h to examine the sustainability of ERK signaling. Nontreated control (cells without NGF and any test compound) and NGF control (cells treated with NGF only) were also grown under the same conditions. Following treatment, cells were washed with ice-cold PBS, scraped in ice-cold NP-40 cell lysis buffer containing 150 mM NaCl, 50 mM Tris (pH 8.0), 2 mM EDTA (pH 8.0) and 1% (*v/v*) NP-40, and incubated on ice for 15 min. The cell lysate was collected by centrifugation (8,000 *g* for 15 min) at 4 °C, and the protein concentration was determined using a BCA kit with bovine serum albumin as a standard.

After boiling for 5 min, the cell lysate (20 µg) was separated on 10% SDS-PAGE and then transferred onto the polyvinylidene difluoride (PVDF) membrane on an iBlot dry blotting system (Invitrogen, Paisley, UK). Nonspecific reactivity was blocked by Bullet Blocking One solution (Nacalai Tesque Inc., Kyoto, Japan) at room temperature for 5 min with shaking. Blots were incubated with the appropriate antibodies: anti-phospho-TrkA (Tyr490) (1:2000), anti-phospho-p44/p42 MAPK (ERK1/2) (Thr202/Tyr204) (1:1000) (Cell Signalling Technology, Danvers, MA, USA), anti-β-actin (1:5000), anti-CREB (1:1000), anti-phospho-CREB (Ser133) (1:1000) (Abcam, Cambridge, UK), and anti-ERK (pan-ERK) (1:5000) (BD, Franklin Lakes, NJ, USA) overnight at 4 °C. After three washes with Tris Buffered Saline containing 0.1% Tween-20 (TBST), the blots were incubated with appropriate horseradish peroxidase-conjugated secondary antibodies (1:5000) (Abcam, Cambridge, UK) for 1 h. The blots were washed with TBST, and the proteins were detected by Amersham ECL Select Western Blotting Detection Reagent (GE Healthcare, Chicago, Illinois, USA) according to the manufacturer’s instructions, and then the chemiluminescence signal was visualized using ChemiDoc Imager (Bio-Rad Laboratories, Hercules, CA, USA)and quantified using Image Lab software, version 6.0.1 (Bio-Rad Laboratories, Hercules, CA, USA).

### 2.5. Western Blot Analysis of AhR Receptor

PC12 cells (50,000 cells/mL per well) were cultured in collagen type IV-coated 24-well culture plates, as mentioned. The cells were then exposed to the antagonists of AhR receptor (ANF and CH223191) at a final concentration of 1 µM for 1 h prior to the treatment with NGF (25 ng/mL) and the test compounds (reference control, ILA or tryptophan at a final concentration of 100 nM) for five consecutive days. Control cells without ANF and CH223191 were also grown and treated under the same conditions. The medium and test compounds were replaced after three days. Following treatment, cell lysate containing 20 µg of total protein were then collected, separated on 8% SDS-PAGE and transferred onto the PVDF membrane on an iBlot dry blotting system. The blots were blocked in Bullet Blocking One solution at room temperature for 5 min with shaking, followed by overnight incubation at 4 °C with the appropriate antibodies: anti-aryl hydrocarbon receptor antibody (1:1000) and anti-β-actin (1:5000) (Abcam, Cambridge, UK). Next, the blots were washed thrice with TBST and incubated with appropriate horseradish peroxidase-conjugated secondary antibodies (1:5000) (Abcam, Cambridge, UK) for 1 h. Signals were developed by Amersham ECL Select Western Blotting Detection Reagent (GE Healthcare, Chicago, Illinois, USA), visualized using ChemiDoc Imager (Bio-Rad Laboratories, Hercules, CA, USA) and quantified using the Image Lab software, version 6.0.1 (Bio-Rad Laboratories, Hercules, CA, USA).

### 2.6. Analysis of AchE Activity upon Pretreatment with AhR Antagonists

PC12 cells (10,000 cells/mL per well) were cultured in collagen type IV-coated 24-well culture plates, as mentioned. The cells were then exposed to the antagonists of AhR receptor (ANF and CH223191) at a final concentration of 1 µM for 1 h prior to the treatment with NGF (25 ng/mL) and the test compounds (reference control, ILA or tryptophan at a final concentration of 100 nM) for five consecutive days. Control cells without ANF and CH223191 were also grown and treated under the same conditions. The medium and test compounds were replaced after three days. Following treatment, the cells were lysed with ice-cold NP-40 cell lysis buffer. AchE activity of the cell lysates was then determined by the Amplite Fluorimetric Acetylcholinesterase Assay Kit (AAT Bioquest, Sunnyvale, CA, USA) according to the manufacturer’s instructions. The AchE activity was determined as described above.

### 2.7. Statistical Analyses

Results are presented as the mean with standard deviation. All statistical analyses were performed using IBM SPSS Statistics, version 22.0, statistical software package (IBM Corp., Armonk, NY, USA). Statistical significance of differences between each treatment group was analyzed by one-way ANOVA with Tukey’s post hoc tests. Intragroup differences were compared by an independent Student’s t-test. A value of *p* < 0.05 was considered to be statistically significant.

## 3. Results

### 3.1. Effects of ILA on Neurite Outgrowth of PC12 Cells

PC12 cells were treated with ILA at concentrations ranging from 1 nM to 1 µM to examine the dose-response. ILA did not morphologically affect neurite outgrowth of PC12 cells in the absence of NGF (data not shown). However, when in the presence of 25 ng/mL NGF, it was found that ILA promoted neurite outgrowth of PC12 cells in a dose-independent manner (Figure 1A). Significant higher neurite outgrowth (*p* < 0.05) was observed in cells treated with ILA at 100 nM, recording a maximal activity. Specifically, at the maximal effective concentration of 100 nM, the percentage of neurite-bearing cells for cells treated with 100 nM ILA reached 24.74 ± 2.19%, which was significantly higher (*p* < 0.05) than those of the reference control (19.78 ± 2.11%), tryptophan (16.02 ± 4.98%) and NGF control (17.11 ± 1.99%) respectively. In addition, significantly higher AchE activity (*p* < 0.01) was also observed in cells treated with ILA at 100 nM (Figure 1B). No significant increases in the percentage of neurite-bearing cells and AchE activity were observed in cells treated with reference control and tryptophan as compared to NGF control. As shown in Figure 1C, prominent neurite outgrowth was elicited by ILA at 100 nM, indicating ILA could potentially enhance NGF-induced neurite outgrowth in PC12 cells.

### 3.2. Effects of ILA on the Ras/ERK Pathway

We next investigated whether TrkA receptor and extracellular signal-regulated kinase 1/2 (ERK1/2) activation are involved in the NGF-induced neurite outgrowth enhanced by ILA in PC12 cells (Figure 2 and Appendix A). Treatment of PC12 cells with ILA (100 nM) in the presence of NGF (25 ng/mL) induced the phosphorylation of TrkA for 24 h (Figure 2A). The relative phosphorylation level of TrkA in ILA-treated cells (1.92 ± 0.15) was higher than the NGF control (1.53 ± 0.17), although no significant difference was observed. Meanwhile, the treatment of PC12 cells with the reference control also tended to increase the phosphorylation of TrkA (1.82 ± 0.09), but no significant difference was observed when comparing to the NGF control. In contrast, the treatment of PC12 cells with tryptophan had no induction effect on the phosphorylation of TrkA (1.62 ± 0.23).

In addition, ILA (*p* < 0.01) strongly induced the phosphorylation of ERK1 (44 kDa) and ERK2 (42 kDa) (Thr202/Tyr204) after 24 h treatment (Figure 2B). The phosphorylation level of ERK1/2 in ILA-treated cells reached 1.98 ± 0.11, and was comparable with the reference control, where a significant increase (*p* < 0.05) in the phosphorylation level of ERK1/2 (1.85 ± 0.073) was also observed upon treatment. Nevertheless, a significant increase (*p* < 0.05) in the phosphorylation level of ERK1/2 was also observed in tryptophan-treated cells (1.75 ± 0.06; *p* < 0.05). These results indicate that ILA could potentially enhance NGF-induced ERK signaling and promote NGF-induced neuronal differentiation in PC12 cells.

We continued to investigate the possible involvement of CREB (cAMP response element-binding protein) in indole derivatives-enhanced NGF-induced neurite outgrowth of PC12 cells. As shown in Figure 2C, the treatment of PC12 cells with ILA at 100 nM significantly increased the phosphorylation of CREB (*p* < 0.05) for 24 h. The phosphorylation level of CREB in ILA-treated PC12 cells reached 2.57 ± 0.48. In contrast, the treatment of PC12 cells with the reference control and tryptophan did not significantly induce CREB phosphorylation as compared to NGF control. These results strongly indicate that the indole derivative, ILA, induced TrkA and ERK1/2 phosphorylation and, in turn, activated CREB transcription in the process of NGF-induced neurite outgrowth of PC12 cells.

### 3.3. Potential Role of ILA as AhR Ligand

We further investigated the potential role of ILA as AhR ligand by western blot analysis and the possible involvement of AhR signaling in the process of neurite outgrowth by the measurement of AchE activity. Significant increases in the relative expression of AhR and the AchE activity upon treatment with ILA in the null PC12 cells were observed (Figure 3, Figure 4 and Appendix A). The results showed that ILA could potentially act as AhR ligand and stimulate the protein expression of AhR for NGF-induced neuronal differentiation in PC12 cells.

Despite certain differences in susceptibility to the AhR antagonists (ANF and CH223191), the effects of ILA on AhR activation were abrogated in the presence of ANF and/or CH223191. As shown in Figure 3, pretreatment of ANF (1 µM) significantly inhibited ILA-induced AhR activity (*p* < 0.05) in PC12 cells, while the AchE activity of ILA-treated cells also tended to decrease. Similarly, ILA-induced AhR activation and AchE activity were also significantly inhibited by CH223191 (1 µM) pretreatment (Figure 4), indicating the specific involvement of AhR in the process of ILA-induced neuronal differentiation. Neither ANF nor CH223191 affected the activity of tryptophan on AhR activation and AchE activity, with the exception in the AchE activity of ANF-pretreated cells.

## 4. Discussion

Studies have shown that indole derivatives produced by gut commensal bacteria from dietary tryptophan could act as vital signaling molecules in intestinal homeostasis, and may be involved in gut–brain communication [2,8,9]. A recent study showed that tryptophan-derived indole compounds have great potential in neuronal health [4]. We report here that tryptophan-derived ILA, which is a unique metabolite produced by the predominant infant gut commensal *Bifidobacterium* spp. [19,20,21,22], potentiated NGF-induced neurite outgrowth in PC12 cells. We showed that the extent of neurite outgrowth could be dose-dependently regulated by ILA. We previously found that ILA was the only tryptophan metabolite produced in bifidobacteria culture supernatants, and its level was substantially higher in bifidobacterial species naturally inhabiting the infant intestines (infant-type HRB). This implies that the beneficial roles of infant-type HRB in the infant’s gut could be attributed to the production of ILA, wherein ILA might be involved in the bifidobacterial host–microbiota crosstalk. Although the exact association between ILA and the health-promoting functions of bifidobacteria, particularly in infant health, remains to be elucidated, our present findings suggest that ILA could be a critical microbial signaling molecule in neuronal developmental processes.

In this study, because of the relative difficulty of studying neuronal differentiation, signaling and other neurobiological events, we made use of PC12 cells as a model system for NGF-induced neurite formation. The well-known neurotrophic factor, NGF, acts on cultured PC12 cells and induces neurite outgrowth that gives rise to sympathetic-like neurons [25]. In response to NGF, PC12 cells cease proliferation, extend long branching neurites, become electrically excitable and express neuronal markers such as increased AchE activity [25,27]. The extent of neuronal differentiation in PC12 cells is, therefore, typically evaluated by counting the number of neurite-bearing cells or examining the AchE activity. Interestingly, our data showed that ILA modestly promoted NGF-induced neurite outgrowth and increased AchE activity in PC12 cells, with a maximal activity at 100 nM. To the best of our knowledge, this is the first report describing the neuro-promoting effects of tryptophan-derived indole compound.

Next, we investigated the possible molecular mechanism of ILA on NGF-induced PC12 cell differentiation. It has been well reported that the extracellular signal-regulated kinase 1/2 (ERK1/2) is the classic signaling cascade often used by various general neurotrophic factors such as NGF for stimulating the acquisition of neuronal phenotypes in chromaffin-like (undifferentiated) PC12 cells [29,30]. In this study, ILA promoted a persistent ERK1/2 activation for NGF-induced neurite outgrowth of PC12 cells. The levels of phosphorylated ERK1/2 proteins significantly increased for 24 h. In PC12 cells, sustained activation of ERK by NGF is required for the induction of neuronal differentiation [30,31]. Studies have shown that only long-term ERK signaling can lead to sufficient protein expression [32,33]. Therefore, in this study, we determined the phosphorylation of the ERK pathway for 24 h. Several tryptophan-related molecules such as indole-3-carbinol [34] and melatonin [35], as well as dietary-derived small molecules such as green tea polyphenols [36], resveratrol [37], and luteolin [38], have also been reported to possess ERK-activating action. In addition, the phosphorylation levels of a specific receptor tyrosine kinase (TrkA) were also greatly elevated in ILA-treated PC12 cells. It has been reported that NGF interacts with TrkA and leads to neurite outgrowth and neuronal differentiation through phosphorylation of ERKs in PC12 cells [39]. In a word, it can be deduced that ILA potentially promoted the NGF-induced neurite outgrowth of PC12 cells by activating the classical ERK1/2 cascade via TrkA receptor.

Furthermore, it is evident that the ERK signaling system is associated with protein translational regulation in the context of long-term potentiation (synaptic plasticity and memory) and converge on the phosphorylation of cAMP response element-binding protein (CREB) at Ser133 [40]. Previous studies have shown that CREB, a ubiquitous transcription factor in neuronal cells, plays critical roles in neuronal differentiation processes and cognitive function [41]. In response to signals that induce neurite outgrowth, Ser133-phosphorylated CREB protein stimulates the expression of numerous target proteins containing cAMP response elements (CREs) in the promoters [41]. In this study, the levels of Ser133-phosphorylated CREB protein were significantly enhanced by ILA. With regard to the intracellular signaling associated with indole derivative treatment for neuronal differentiation in PC12 cells, the present results clearly demonstrate that ILA potentially promoted NGF-induced neuronal elongation in PC12 cells through the Ras/ERK signaling pathway. Future studies are needed to elucidate the detailed mechanism underlying ILA-induced ERK-dependent neuritogenesis.

On the other hand, it is increasingly apparent that bacterial tryptophan-derived indole compounds, including ILA, act as ligands of aryl hydrocarbon receptor (AhR) [10,11]. AhR is a ligand-activated transcription factor that regulates many vital physiological processes, including metabolism, differentiation, development and reproduction [42]. A growing body of studies has demonstrated that AhR activation modulates innate and adaptive immune responses in a ligand-specific fashion [43]. Specifically, indole derivative-induced AhR activation has been reported to be one potential way that bacteria contribute to mucosal homeostasis. For instance, indole-3-aldehyde (IAld) produced by *Lactobacillus* spp. was found to be involved in AhR-mediated regulation of in IL-22 mucosal homeostasis [11]. In addition, the treatment of mice with three *Lactobacillus* strains capable of metabolizing tryptophan attenuated intestinal inflammation via AhR activation, as the effects were reversed by an AhR antagonist [44]. Therefore, it is speculated that ILA may act as AhR agonists and enhance NGF-induced neurite outgrowth of PC12 cells via AhR signaling.

In an attempt to unravel the possible role of ILA as AhR ligand for enhancement of neurite outgrowth, the protein expression of AhR was determined upon treatment with NGF and ILA for five consecutive days. Significant increases in the relative expression of AhR were observed in PC12 cells treated with ILA, confirming the role of ILA as AhR agonist. Interestingly, such an elevation of AhR expression was associated with increased AchE activity in ILA-treated PC12 cells, indicating AhR activation induced by ILA evokes neurite outgrowth in PC12 cells. Previous reports have revealed that AhR is involved in neuronal differentiation [45]. The expression of AhR has been characterized in murine models where AhR is expressed mainly in the neurons of the cortex, the cerebellum, the hippocampus and the olfactory bulb [46]. More specifically, AhR is expressed in the early developmental stages. The mRNA of AhR is detected in neural progenitor cells in the hippocampus and in early brain structures in mice, and its expression increases throughout development [47,48].

In this study, the specificity of AhR activation by ILA was examined using the receptor antagonists ANF and CH223191. It has been reported that ANF and CH223191 compete with AhR agonists in binding to the AhR, and inhibit the ability of AhR agonists to activate the AhR signal transduction [49,50]. We found that pretreatment with the AhR antagonists greatly inhibited the effects of ILA on AhR activation and AchE activity in PC12 cells, thus indicating specific AhR-mediated signaling in ILA-enhanced NGF-induced neurite outgrowth of PC12 cells. Previous reports have shown that metabolites derived from dietary tryptophan by the gut microbiota activate AhR signaling in astrocytes and control inflammatory processes triggered by astrocytes [51,52,53]. Supplementation with the tryptophan metabolites indole, indoxyl-3-sulfate (I3S), IPA and IAld, or the bacterial enzyme tryptophanase reduced central nervous system inflammation in antibiotic-treated mice [52]. Combined with our present findings, it is likely that tryptophan-derived ILA could potentially activate AhR signaling in PC12 cells and promote neuronal differentiation. Future studies are needed to elucidate the exact role AhR in the processes of neuronal differentiation and the underlying mechanisms of ILA in AhR-mediated neurite outgrowth.

## 5. Conclusions

In conclusion, the present study revealed that tryptophan-derived ILA potentiated NGF-induced neurite outgrowth in PC12 cells through the Ras/ERK signaling pathway. ILA was found to promote NGF-induced neurite outgrowth of PC12 cells, possibly by increasing the phosphorylation level of TrkA receptor and ERK1/2, CREB in Ras/ERK signaling cascade at a concentration of 100 nM. In addition, ILA could also potentially act as AhR agonist and stimulate AhR-mediated neurite outgrowth in PC12 cells. To the best of our knowledge, we are the first to discover the potential role of tryptophan-derived indole compounds on neuronal differentiation. The discovery of ILA as an AhR agonist may provide new insights into how microbial tryptophan metabolites mediate in gut–brain communication. These new findings lay the ground for the potential involvement of the indole derivative, ILA, which is prevalently produced by infant gut commensal bifidobacteria, as the key mediator in host–microbial crosstalk and the neuronal developmental processes. Additional evaluation of the molecular mechanisms of action of ILA may provide clues by which to gain a better understanding of the specific involvement of indole derivatives in the communication between the commensal bifidobacteria, gut microbiota and the host for neuronal development.

## Figures and Tables

**Figure 1 microorganisms-08-00398-f001:**
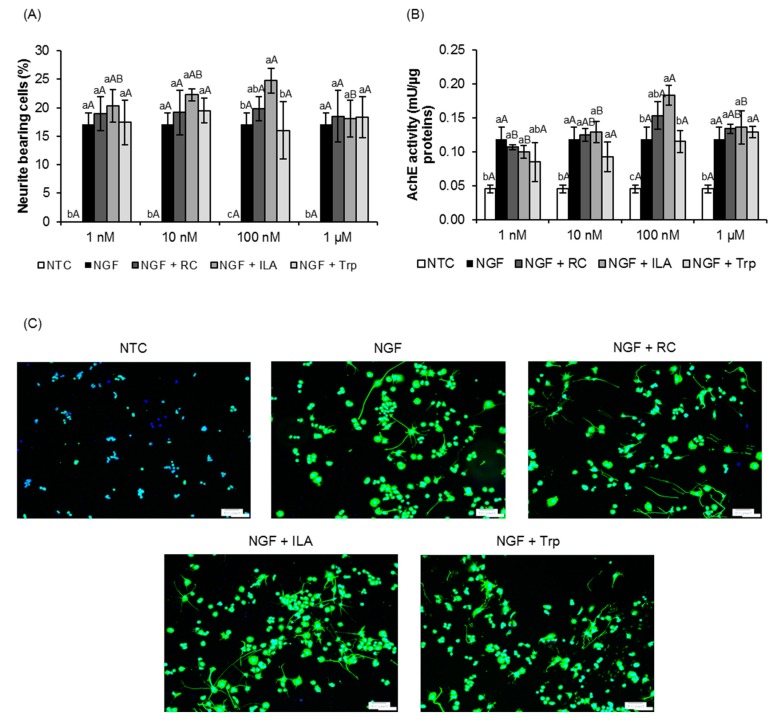
Effects of indole-3-lactic acid (ILA) on nerve growth factor (NGF)-induced neurite outgrowth of PC12 cells. PC12 cells were treated with NGF (25 ng/mL) and the test compounds (RC, ILA or Trp at 1 nM, 10 nM, 100 nM, and 1 µM) for five consecutive days. (**A**) Percentage of neurite-bearing cells in PC12 cells. (**B**) Acetylcholinesterase (AchE) activity in PC12 cells. (**C**) Images of βIII-tubulin (green) immunostaining of PC12 cells treated with NGF (25 ng/mL) and the test compounds (RC, ILa or Trp at 100 nM) for five consecutive days at a magnification of ×100. Scale bars: 100 µm. Nuclei were counterstained with DAPI (blue). The data represent the mean ± SD of three replicates. One-way ANOVA with Tukey’s post hoc tests were used for statistical analysis. Means with different lowercase letters are significantly different between test groups (*p* < 0.05). Means with different uppercase letters are significantly different within the same test group (*p* < 0.05). NGF-treated control. NTC, nontreated control; papaverine hydrochloride was used as a reference control (RC); tryptophan (Trp) was used as a baseline compound.

**Figure 2 microorganisms-08-00398-f002:**
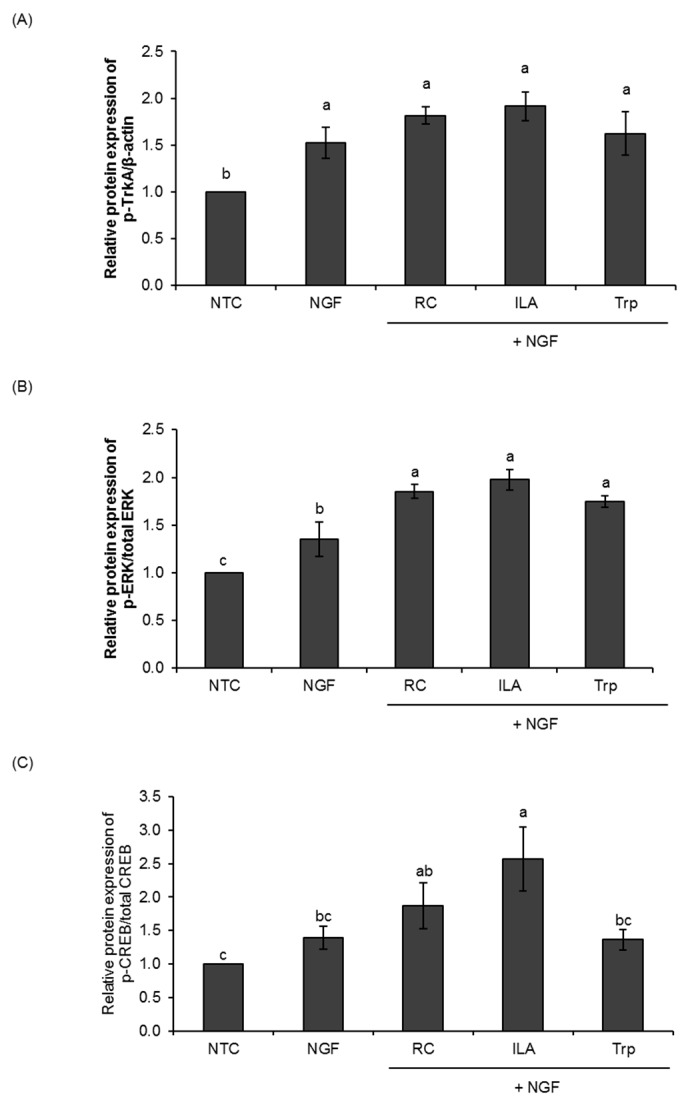
Effects of indole-3-lactic acid (ILA) on the phosphorylation of TrkA, ERK1/2, and CREB in PC12 cells. Phosphorylation of (**A**) TrkA, (**B**) ERK1/2, and (**C**) CREB in PC12 cells treated for 24 h with NGF (25 ng/mL) and RC, ILA, or Trp (100 nM) was detected using western blot analysis. The data represent the mean ± SD of three replicates. Means with different lowercase letters are significantly different between test groups (*p* < 0.05) by one-way ANOVA with Tukey’s post hoc tests. NTC, nontreated control; Papavarine hydrochloride was used as a reference control (RC); Tryptophan (Trp) was used as a baseline compound.

**Figure 3 microorganisms-08-00398-f003:**
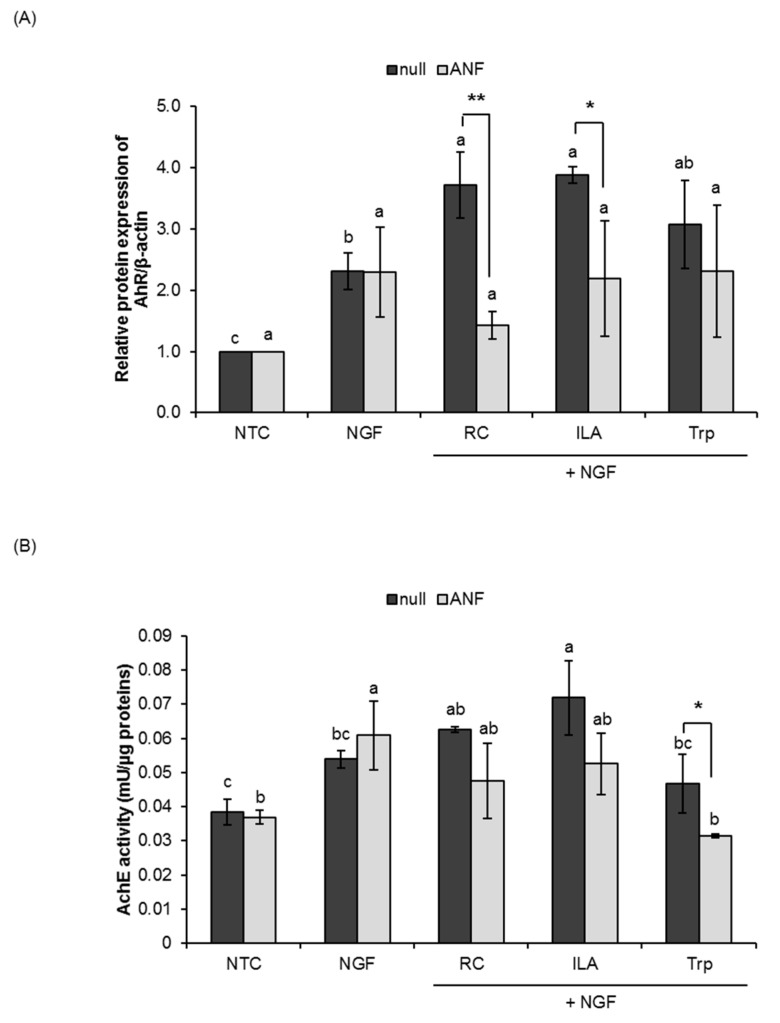
Effects of indole-3-lactic acid (ILA) on the aryl hydrocarbon receptor (AhR) in PC12 cells. PC12 cells were pretreated with the AhR antagonist, α-naphthoflavone (ANF; 1 µM), for 1 h and treated for five consecutive days with NGF (25 ng/mL) and RC, ILA or Trp (100 nM). Nonpretreated PC12 cells served as a null control. (**A**) AhR protein (95 kDa) in PC12 cells was detected by Western blot analysis using a monoclonal antibody specific for AhR. The corresponding β-actin blot served as a loading control. (**B**) Acetylcholinesterase (AchE) activity in PC12 cells. The data represent the mean ± SD of three replicates. ^*^
*p* < 0.05, ^**^
*p* < 0.01 for intragroup differences compared by an independent Student’s t-test. Means with different lowercase letters are significantly different between test groups (*p* < 0.05) by one-way ANOVA with Tukey’s post hoc tests. NTC, nontreated control; Papavarine hydrochloride was used as a reference control (RC); Tryptophan (Trp) was used as a baseline compound.

**Figure 4 microorganisms-08-00398-f004:**
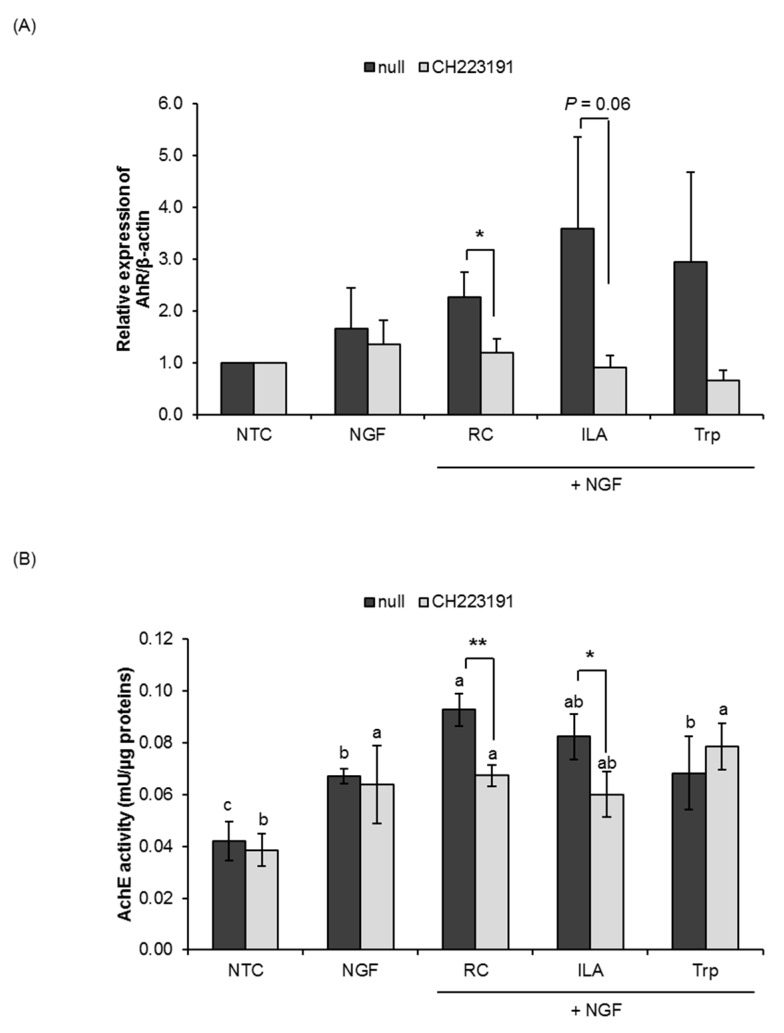
Effects of indole-3-lactic acid (ILA) on the aryl hydrocarbon receptor (AhR) in PC12 cells. PC12 cells were pretreated with the AhR antagonist, CH223191 (1 µM), for 1 h and treated for five consecutive days with NGF (25 ng/mL) and RC, ILA or Trp (100 nM). Nonpretreated PC12 cells served as a null control. (**A**) AhR protein (95 kDa) in PC12 cells was detected by Western blot analysis using a monoclonal antibody specific for AhR. The corresponding β-actin blot served as a loading control. (**B**) Acetylcholinesterase (AchE) activity in PC12 cells. The data represent the mean ± SD of three replicates. ^*^
*p* < 0.05, ^**^
*p* < 0.01 for intragroup differences compared by an independent Student’s *t*-test. Means with different lowercase letters are significantly different between test groups (*p* < 0.05) by one-way ANOVA with Tukey’s post hoc tests. NTC, nontreated control; Papavarine hydrochloride was used as a reference control (RC); Tryptophan (Trp) was used as a baseline compound.

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
