# Peer review of "Potential Effects of Indole-3-Lactic Acid, a Metabolite of Human Bifidobacteria, on NGF-Induced Neurite Outgrowth in PC12 Cells"

_microorganisms, 2020, doi:10.3390/microorganisms8030398_

Round 1

Reviewer 1 Report

The results of this paper indicate that indole-3-lactic acid (ILA), a metabolite of human infant-type bifidobacteria, enhanced NGF-induced neurite outgrowth and acetylcholinesterase activity of PC12 cells, suggesting a potential role of ILA in a neuronal development process. ILA also increased NGF-induced phosphorylation of TrkA, ERK, and CREB and AhR protein levels. The promoting effects of ILA on both NGF-induced AhR protein levels and acetylcholinesterase activity were suppressed by an AhR antagonist. These results suggest that ILA enhances NGF-induced differentiation of PC12 cells through mechanisms involving a TrkA-ERK-CREB signaling pathway and AhR signaling.

   The paper uses appropriate methodology and well-organized, and most of the data seem reliable. The study provides an important contribution to our knowledge regarding the roles of bifidobacteria in the host neuronal development. However, I have some concerns and suggestions to strengthen authors’ argument.

  1. Phosphorylations of TrkA, ERK and CREB are early events and peak after minutes of NGF exposure. The reason why those phosphorylations were determined 24 h after NGF addition needs to be explained in the text.
  2. AhR Western blot data with no AhR antagonist in Figures 3A and 4A are the results of the same treatments. Why are the mean values and SDs so different between the two figures?
  3. Related to my comment 2, the data of Figures 1-4 are expressed as mean ± SD of three independent experiments or mean ± SD of three wells in a representative experiment?
  4. The authors claim the involvement of AhR in the enhancing effect ILA on NGF-induced neuronal differentiation (Fig. 4B). Do the AhR antagonists affect phosphorylations of TrkA, ERK and CREB induced by NGF plus ILA?
  5. Lines 265-266. "The results showed that ILA could potentially act as AhR ligand, with a significant increase in the relative expression of AhR upon the treatment (Figures 3, 4 and S2)." More understandable explanation why an AhR ligand induces AhR protein level may be required.
  6. Lines 245-246. "These results indicate that ERK signalling is involved in the ILA-enhanced NGF-induced neuronal differentiation of PC12 cells." Careful statement is required because the authors' data only show that ILA enhanced both ERK phosphorylation and neurite outgrowth induced by NGF, and thus no evidence is provided here for the involvement of ERK signaling in ILA-enhanced NGF-induced neuronal differentiation.
  7. Minor mistakes: b-actin→β-actin (vertical axes of Figures 2A, 3A, and 4A); 1nM, 10nM, 100nM, and 1μM→1 nM, 10 nM, 100 nM, and 1 μM (horizontal axes of Figures 1A and 1B).

Author Response

Point 1:  Phosphorylations of TrkA, ERK and CREB are early events and peak after minutes of NGF exposure. The reason why those phosphorylations were determined 24 h after NGF addition needs to be explained in the text.

Response 1: In PC12 cells, sustained activation of ERKs by NGF is required for the induction of neuronal differentiation. Study has shown that only long-term ERK signalling can lead to sufficient protein expression. Therefore, in this study, we determined the phosphorylation of ERK pathway for 24 h. The explanations with appropriate references were added in the text (Line 150 and Line 337-340).

Point 2: AhR Western blot data with no AhR antagonist in Figures 3A and 4A are the results of the same treatments. Why are the mean values and SDs so different between the two figures?

Response 2: Please note that the data with no AhR antagonist in Figures 3A and 4A are different results of independent treatments. Therefore, the mean values and SDs are different between the two figures.

Point 3: Related to my comment 2, the data of Figures 1-4 are expressed as mean ± SD of three independent experiments or mean ± SD of three wells in a representative experiment?

Response 3: The data of Figures 1-4 are expressed as mean ± SD of three independent replicates.

Point 4: The authors claim the involvement of AhR in the enhancing effect ILA on NGF-induced neuronal differentiation (Fig. 4B). Do the AhR antagonists affect phosphorylations of TrkA, ERK and CREB induced by NGF plus ILA?

Response 4: Thank you for pointing out this. To elucidate the possible involvement of AhR in ILA-enhanced NGF-induced neurite outgrowth, the authors only examine the direct effect of AhR antagonists on the known differentiation phenotype of PC12 cells (AchE activity) but not the phosphorylation of TrkA, ERK and CREB.

Point 5: Lines 265-266. "The results showed that ILA could potentially act as AhR ligand, with a significant increase in the relative expression of AhR upon the treatment (Figures 3, 4 and S2)." More understandable explanation why an AhR ligand induces AhR protein level may be required.

Response 5: A more understandable explanation was added in Line 270-273.

Point 6: Lines 245-246. "These results indicate that ERK signalling is involved in the ILA-enhanced NGF-induced neuronal differentiation of PC12 cells." Careful statement is required because the authors' data only show that ILA enhanced both ERK phosphorylation and neurite outgrowth induced by NGF, and thus no evidence is provided here for the involvement of ERK signaling in ILA-enhanced NGF-induced neuronal differentiation.

Response 6: This sentence (Line 247-249) was amended accordingly to reflect that ILA only enhanced both ERK phosphorylation and neurite outgrowth induced by NGF.

Point 7: Minor mistakes: b-actin→β-actin (vertical axes of Figures 2A, 3A, and 4A); 1nM, 10nM, 100nM, and 1μM→1 nM, 10 nM, 100 nM, and 1 μM (horizontal axes of Figures 1A and 1B).

Response 7: These minor mistakes were amended accordingly.

Reviewer 2 Report

Indole-3-lactic acid (ILA), a tryptophan metabolite of breast milk, has previously been shown to be an anti-inflammatory molecule. ILA reduces the response of interleukin-8 (IL-8). It interacts with the transcription factor of the aryl hydrocarbon receptor (AHR) and prevents the transcription of the inflammatory cytokine IL-8.

The authors attempt to emphasize the importance of higher levels of ILA, suggesting that infant-type HRB may play a specific role in crosstalk of the host microbiota, producing ILA in infants and in regulating neuronal differentiation. By adding ILA and NGF to PC12 cells, they control neurite outgrowth and acetylcholinesterase activity. They found that ILA significantly enhances NGF-induced basal expansion of neurites. In addition, ILA induces TrkA, ERK1 / 2, and CREB receptor phosphorylation in neurons. Therefore, the authors conclude that ILA enhances neurite outgrowth along the Ras / ERK pathway.

I am not sure that the data can fully support the conclusions. In particular:

The images shown in Figure 1 are not representative since the quantification in Figure 1 was not shown. There was only a slight increase between + NGF and + NGF + Trp. However, in the images, the green intensity between + NGF and + NGF + ILA was the same, while in the control, it was also the same.

The increase in ERK1 / 2 phosphorylation after induction of various metabolites does not change, as I see in S1B. Can authors inhibit the phosphorylation of ERK1 / 2 (pharmacologically or by knockdown kinases) and see if the induced difference between treatment with NGF alone is abolished?

In some experiments, I would like to see real-time PCR for a deeper understanding of the effect of ILA on neuronal differentiation
